# Lights and Shadows on the Cancer Multi-Target Inhibitor Rigosertib (ON-01910.Na)

**DOI:** 10.3390/pharmaceutics15041232

**Published:** 2023-04-13

**Authors:** Ana Monfort-Vengut, Guillermo de Cárcer

**Affiliations:** Cell Cycle and Cancer Biomarkers Group, Instituto de Investigaciones Biomédicas Alberto Sols (IIBM) CSIC-UAM, 28029 Madrid, Spain

**Keywords:** rigosertib, ON-01910.Na, cancer therapy, kinase allosteric inhibition, Plk1, Ras, JNK, PI3K/Akt, microtubule-destabilizing agents, stress-signaling pathways

## Abstract

Rigosertib (ON-01910.Na) is a small-molecule member of the novel synthetic benzyl-styryl-sulfonate family. It is currently in phase III clinical trials for several myelodysplastic syndromes and leukemias and is therefore close to clinical translation. The clinical progress of rigosertib has been hampered by a lack of understanding of its mechanism of action, as it is currently considered a multi-target inhibitor. Rigosertib was first described as an inhibitor of the mitotic master regulator Polo-like kinase 1 (Plk1). However, in recent years, some studies have shown that rigosertib may also interact with the PI3K/Akt pathway, act as a Ras–Raf binding mimetic (altering the Ras signaling pathway), as a microtubule destabilizing agent, or as an activator of a stress-induced phospho-regulatory circuit that ultimately hyperphosphorylates and inactivates Ras signaling effectors. Understanding the mechanism of action of rigosertib has potential clinical implications worth exploring, as it may help to tailor cancer therapies and improve patient outcomes.

## 1. Introduction

Rigosertib (ON-01910.Na) is a non-cyclooxygenase inhibitor sulfone metabolite with the ability to act as a non-ATP competitor kinase inhibitor. Originally described as an inhibitor for the master mitotic kinase Polo-like kinase (Plk1) [1], rigosertib has been reported to have broad activity against a number of different targets, including kinases such as PI3K/Akt or NF-kB complex proteins [2,3]. Interestingly, rigosertib has also been described as a Ras mimetic molecule, inhibiting Ras–Raf binding and, therefore, hampering Ras mitogenic signaling [4]. Recent data from CRISPR genome-wide screens describe rigosertib as a microtubule-depolymerizing agent that leads to cell-cycle alterations that are typical of other microtubule dynamic poisons [5]. It has also been proposed that the induced mitotic stress, upon rigosertib treatment, leads to oxidative stress resulting in a JNK-mediated inhibition of the Ras–MAPK pathway [6]. Overall, rigosertib is highly effective in killing tumoral cells in vitro and in vivo, with cells arrested in mitosis being the most prominent phenotype, although there is abundant data showing that rigosertib effectively stops cell proliferation through various combined effects.

Based on the promising in vitro data, rigosertib was subjected to clinical trials and showed good tolerability in patients with low levels of toxicity. As a result, rigosertib reached phase III trials for second-line treatment of patients with high-risk myelodysplastic syndrome (MDS) [7] and, in combination with gemcitabine, for patients with metastatic pancreatic cancer [8]. Unfortunately, these clinical trials were not successful, as the patients did not show a significant benefit over the standard of care, suggesting a need to find the right sensitivity biomarkers to achieve a therapeutic benefit. In summary, the multi-target activity of rigosertib, which is controversial, makes this drug an interesting molecule to study in order to deeply understand its mechanism of action and, subsequently, to find the biomarkers that can make rigosertib a profitable drug in the treatment of diseases.

This review collects data on the different targets and cell phenotypes associated with rigosertib response and attempts to clarify the targeting controversy raised in recent years. We also discuss some of the previously described mechanisms of resistance to rigosertib and, finally, summarize all the rigosertib clinical trials conducted to date. Overall, our aim is to help the growing number of researchers working with this small compound.

## 2. Development of Rigosertib (ON-01910.Na)

One of the main problems associated with kinase-inhibition-based strategies is the development of drug resistance due to the accumulation of mutations in the ATP binding site of the kinase [9]. The scientific community has focused its efforts on developing inhibitors that target regions outside the ATP binding site of the target kinases and may act as better drug candidates in cancer therapy.

Between 2002 and 2003, Reddy et al. described the synthesis of several small-molecule kinase inhibitors, based on styryl-benzyl-sulfones, that exhibited high antitumor activity and were not ATP-competitors, but allosterically inhibited substrate binding [10,11,12]. Since the cytotoxic activity depends on the nature and position of the styryl aromatic ring, the group synthesized a collection of molecules capable of targeting cancer cells while leaving non-malignant cells unaffected, thus lacking many side effects associated with commonly used chemotherapeutic agents. This advantage opened a therapeutic opportunity window for this collection of new small molecules against cancer.

One of the most analyzed styryl-benzyl-sulfones is rigosertib (ON-01910), commercially known as Estybon^®^. Rigosertib was derived from ON-01940 ((E)-2′,4′,6′-trimethoxystyryl-4-methoxy-3-aminobenzylsulfone) in a two-step synthetic reaction (Figure 1): first, ON-01940 is reacted with a methyl 2-bromo acetate in a mild basic sodium acetate media to generate ON-01500 ((E)-2-methoxy-5-(((2,4,6-trimethoxystyryl)sulfonyl)methyl)aniline). Second, ON-01500 is hydrolyzed with sodium hydroxide in ethanol and dichloromethane, followed by washing with methyl ethyl ketone to obtain ON-01910 (rigosertib, sodium (E)-2-{2-methoxy-5-[(2′,4′,6′-trimethoxystyrylsulfonyl)methyl]phenylamino}acetate [13]. After screening these compounds for activity against tumoral cell lines, the researchers found ON-01910 to be a potent inhibitor of the mitotic kinase Plk1, inducing mitotic arrest of tumor cells. It is worth noting that rigosertib also showed certain affinity to other kinases, such as PDGFR, Abl, Flt-1, CDK1, Plk2, Src, and Fyn, which share common features in their binding sites [1].

After rigosertib was first described, research began to flourish and many studies have provided new insights into this small molecule, describing new mechanisms and putative targets such as the PI3K–Akt pathway [14] the Ras–Raf signaling cascade [4], and microtubule dynamics [5] (Figure 2).

## 3. Described Targets for Rigosertib

Over the years, many studies have been published in attempts to unravel the mechanism of action of rigosertib, to the point where it is now considered a multi-kinase inhibitor. In recent years, rigosertib has also been described as a non-kinase drug, further complicating the understanding of its mechanism of action. The main targets described are the following:

### 3.1. Rigosertib as a Plk1 Inhibitor

Rigosertib has been generally considered a Plk1 inhibitor. Pioneering phosphorylation experiments performed immediately after rigosertib synthesis, combining recombinant Plk1 and CDC25 or Casein as substrates, showed a very efficient rigosertib-mediated Plk1 inhibition with an IC_50_ of 9–10 nM [1]. In parallel, the chemical mechanism of action of rigosertib was defined as allosteric, since increasing concentrations of ATP in the kinase reaction did not alter the inhibition constant. Notably, the allosteric binding of rigosertib to Plk1 has not yet been demonstrated [15]. 

To further evaluate the specificity of rigosertib for Plk1, the authors evaluated the activity of a panel of 29 recombinant kinases and showed that rigosertib was most active against Plk1 and, therefore, likely to be its primary target. However, rigosertib also showed some activity against other kinases, such as PDGF receptor (PDGFR), Abl, Flt-1, CDK1, Plk2, and Src, when tested in recombinant kinase assays. Similarly, when the authors performed cell synchronization assays and tested for Plk1 activation, they showed that rigosertib inhibited Plk1 activity, although total levels of the kinase were not affected. This effect of rigosertib seemed to be specific for Plk1, since there is almost no alteration in CDK1 activity, although other kinases were not tested in these experiments. Since Plk1 is required for tumor cell proliferation, the authors showed that rigosertib induced apoptotic activity against 94 tumor cell lines with a GI_50_ between 50–200 nM, resulting in tumor cell death by mitotic arrest, demonstrating its potential efficacy. Notably, this cytotoxic effect was not observed in non-tumoral cells. More recently, in an effort to evaluate the efficacy of rigosertib in retinoblastoma-derived cells, Ma et al. showed suppression of Cdc25C phosphorylation as a readout for Plk1 activity and concomitant accumulation of CDK1 phosphorylation [16]. They also performed flow cytometry cell-cycle analysis and demonstrated that upon rigosertib treatment there was an arrest in the G2/M phase of the cell cycle, although it was not as strong as that identified in other reports.

Polo-like kinase 1 is a master regulator of cell division, controlling several important processes throughout the cell cycle, including centrosome maturation, spindle assembly, chromosome segregation, and, ultimately, cytokinesis [17]. The chemical inhibition or genetic depletion of Plk1 results in strong mitotic arrest and efficient cell death, making it a bona fide cancer target [18]. Importantly, Plk1 inhibition produces a highly recognizable phenotype, which is the presence of monopolar mitotic spindles [19,20] leading to impaired mitotic progression and subsequent mitotic arrest.

To confirm Plk1 as a substrate for rigosertib, confocal microscopy experiments were performed to examine the cell fate of treated cells. Rigosertib-treated HeLa cells exhibit mitotic spindle aberrations, such as multipolar spindles, chromosome mislocalization in metaphase cells, and centrosome fragmentation [1]. Although, as mentioned above, the chemical inhibition of Plk1 is known to induce mitotic arrest, the observed phenotypes in the HeLa-treated cells did not accurately reflect the classical Plk1 inhibition phenotype, which consists of monopolar spindle organization due to impaired centrosome duplication [21]. Indeed, other studies have shown that rigosertib-treated U2OS do not show alterations in centrosome maturation, a well-known function for Plk1, and present increased levels of gamma-tubulin loading at the centrosomes [22].

As the Plk1 inhibition phenotype is controversial, there are other possible explanations for the mitotic arrest observed with rigosertib treatment. Oussenko et al. described how rigosertib induces a prolonged phosphorylation of RanGAP1-SUMO1, thereby arresting cells in mitosis and ultimately inducing apoptosis in acute lymphoblastic leukemia (MOLT-3), prostate cancer (DU-145), and lymphoma cells (U937) [23]. As it is well known that RanGAP1 is phosphorylated at the onset of mitosis [24], it is difficult to determine whether RanGAP1-SUMO1 phosphorylation is directly due to rigosertib function or, on the other hand, a secondary effect of the mitotic arrest mediated by rigosertib. The authors claimed that RanGAP1-SUMO1 phosphorylation happens in short time after adding rigosertib, thus not having enough time to arrest in mitosis.

In summary, although there is still a tendency to use rigosertib as a Plk1 inhibitor [16,25,26,27], the Plk1 inhibition mechanism for rigosertib is controversial, as there is strong evidence that this may not be the real mechanism and some alternative mechanisms have been described for the mitotic arrest phenotype.

### 3.2. Rigosertib as a PI3K–Akt Pathway Inhibitor

Although most of the initial studies pointed to rigosertib as a Plk1 inhibitor, or more generally as a mitotic inhibitor [14,28,29,30,31,32,33], some reports additionally demonstrated that it disrupted other non-cell-cycle-related signaling pathways. Xu et al. hypothesized that since rigosertib was being used as a kinase inhibitor, it was worth studying the dysregulation of signaling transduction pathways after rigosertib treatment; they tested samples from myelodysplastic syndrome (MDS) patients and found 31 significant upregulated pathways and 14 downregulated pathways. The PI3K–Akt, MAPK, Jak-STAT, Wnt, and Notch signaling pathways were among the most dysregulated in MDS patients, which could point to apoptotic cell death regulation [14].

These phenotypes were also observed in cell lines derived from head and neck squamous cell carcinoma (HNSCC), where PI3K dysregulation is a common event. Authors demonstrated that rigosertib induced toxicity in HNSCC cell lines, inhibiting PI3K/Akt/mTOR pathways in a dose-dependent manner, which ultimately led to the inhibition of cell-cycle progression. It also induced oxidative stress that generated reactive oxygen species (ROS) and activated extracellular-regulated kinases (ERK1/2) and c-Jun NH2-terminal kinase (JNK). They demonstrated that rigosertib triggered the mitochondria translocation of the stress kinase ATF-2, which is also related to apoptosis induction [3]. In contrast, another study done in HNSCC found rigosertib to be an effective treatment, but by a completely different mechanism than PI3K–Akt inhibition [2]. Here, the authors did not observe any alterations in the PI3K pathway, such as the phosphorylation on Akt Ser-473 residue, or Ser-240/244 of the S6 ribosomal protein. On the contrary, a complete G2 arrest is observed, which fits with rigosertib being a mitotic inhibitor.

Overall, the possible inhibition of the PI3K–Akt axis by rigosertib seems to be cell-dependent and often accompanied by mitotic arrest. Therefore, it is difficult to interpret if both mechanisms are autonomous.

### 3.3. Rigosertib as a Ras Mimetic and Ras–Raf–MEK Axis Inhbitor

Up to this point, observations suggested that rigosertib’s mechanism of action involved several alternatives and appeared to be dependent on cell type and genetic background. In 2016, the laboratory of E. Premkumar Reddy described rigosertib as the first Ras–RBD inhibitor, binding to the RBD (Ras-binding domain) of the Ras effectors and impeding Ras binding, therefore blocking the activation of Raf kinase activity and inhibiting Ras–Raf–MEK signaling [4]. The binding of rigosertib to the Raf–RBD domain was elegantly demonstrated by NMR studies, further showing that rigosertib does indeed impede Raf heterodimerization. Notably, the authors claimed that rigosertib specifically inhibits the Ser-388 residue phosphorylation of c-Raf that associates with Plk1 [34], and this may explain the relationship between rigosertib and Plk1 inhibition. Likewise, other studies showed that rigosertib treatment induces cell-cycle arrest and disrupts Ras signaling [35,36]. Finally, the authors also found that rigosertib can bind to the RBDs of Ral-GDS and PI3Ks, which might explain the inhibitory mechanism on PI3K and, therefore, its multi-target capacity.

On the other hand, Ritt et al. demonstrated that rigosertib leads to an inhibition of the Ras/Raf/MEK/ERK signaling axis, but in an indirect way. Rigosertib-induced mitotic arrest generates an oxidative stress that activates the stress cascade mediated by the cJun N-terminal kinases 1/2 (JNK1/2). This stress signaling hyperphosphorylates RasGEF, Sos1, and the Raf proteins, rendering them unresponsive to upstream signals [6]. Notably, this effect of rigosertib can be mimicked by other microtubule poisons such as taxol, suggesting that mitotic arrest is the main inducer of such a stress response, rather than a specific targeting of Ras–MEK signaling by rigosertib. In the same trend, Günther et al. provided more detail on the JNK-mediated mechanism of rigosertib, showing that rigosertib treatment activates p66Shc [37], which is a known effector of the JNK1/2 kinases [38]. Subsequently, p66Shc translocates to the mitochondria, where it induces ROS production [39].

Following the description of the Ras–Raf inhibitory activity of rigosertib, there has been a growing collection of articles exploiting this evidence. Rahmani et al. investigated the mechanism of rigosertib in colorectal cancer and confirmed that it inhibited cell proliferation and the cell cycle and that it was dependent on mutations in K-Ras [40], which also elevates ROS. More interestingly, they showed that rigosertib regulates the expression of CD31, an angiogenic marker, which opens an alternative for rigosertib as an antiangiogenic drug.

Rigosertib was also used as a canonical Ras–Raf signaling inhibitor with the intention to evaluate the impact of Ras signaling during the immune checkpoint blockade (ICB) in metastatic melanoma. The efficacy of rigosertib in melanoma is dependent on the activity of CD40, a receptor molecule of the TNF receptor family that is normally expressed on antigen-presenting cells [41]. Rigosertib Ras/Raf/PI3K inhibition promotes CD40 expression that induces immunogenic cell death and enhances the response to immune checkpoint inhibitors, thus conferring immunogenicity to rigosertib [42]. Whether this effect on CD40 is due to Ras–Raf signaling or to the dependence of CD40 function upon ROS induction [43,44] is not yet clear. Finally, in a non-cancer related study, Wang et al. studied lipopolysaccharide-induced sepsis, using rigosertib as a MEK1-ERK signaling inhibitor, concluding that it prevented the production of proinflammatory cytokines induced by LPS by disrupting the activation of the MEK1-ERK signaling axis [45].

Overall, although strong data show that rigosertib interferes with Ras–Raf–MEK signaling, the precise mechanism by which this effect is mediated is still unclear, and it is also unclear whether it is a direct molecular impact on Ras–Raf molecules or indirectly induced by cytotoxic stress mediated by ROS induction and stress kinase (JNK) signaling.

### 3.4. Rigosertib as a Tubulin Polymerization Destabilizer

A common cellular phenotype upon rigosertib treatment, observed by many different groups, is the G2/M arrest accompanied by mitotic aberrations. As mentioned above, this was originally assumed to be dependent on the Plk1 inhibition. However, there are examples providing evidence that rigosertib might be a microtubule dynamics inhibitor and this might be also responsible for the mentioned G2/M arrest.

Hyoda et al. showed that rigosertib induced the phosphorylation of histone H2AX and led to a DNA damage-induced G2/M arrest in HL-60, Jurkat, Ramos, and MDS-L cell lines. Rigosertib caused an abnormal localization of Aurora A kinase (AurKA), suggesting that it disrupts the spindle assembly machinery, and they finally concluded that the rigosertib effect had a possible involvement with genes related to microtubule kinetics [28]. More recently, the efficacy and mechanism of rigosertib have been tested in rhabdomyosarcoma (RMS) and neuroblastoma (NBs) cell lines. Here, the authors showed that the main cytotoxic effect is due to an efficient cell arrest in mitosis, mainly by interfering with mitotic spindle assembly, which leads to cell death [46,47]. In fact, the authors have shown a similar behavior of rigosertib with respect to other Vinca alkaloids, although it is better tolerated. Interestingly, these studies also provided evidence that rigosertib does not induce cell death by inhibiting the Ras pathway in either RMS or NB cells, demonstrating that the mechanism of action of rigosertib may be cell-type dependent.

An explanation for the rigosertib-mediated mitotic arrest was recently described by Jost et al., who claimed that rigosertib is indeed a microtubule-destabilizing agent [5]. Their study aimed to identify novel rigosertib response-associated biomarkers by performing a combined CRISPRi/a (inactivation/activation) genome-wide screen using the myeloid leukemia cell line K562. The authors demonstrated that rigosertib causes spindle and mitotic defects because it binds to αβ-tubulin heterodimers, in a region similar to that of to other microtubule destabilizers, such as colchicine. This binding of rigosertib to tubulin alters microtubule dynamics both in vitro and in vivo. Moreover, the authors used structural analysis to depict the residues involved in the binding of rigosertib to β-tubulin, with Lys-240 being critical for this interaction. Interestingly, the mutation of this Lys-240 to phenylalanine (L240F) confers resistance to rigosertib in K562 cells, confirming that the rigosertib effect is mainly mediated by altering microtubule dynamics and, subsequently, mitotic alterations.

An important aspect to consider in these results is that rigosertib is synthesized from ON-01500 (Figure 1), which is a microtubule poison. In fact, this feature opened a dispute when the Reddy group published a contrary article stating that the microtubule-destabilizing effect is most likely due to the presence of the late intermediate impurity ON-01500 in commercial rigosertib formulations, which has a potent tubulin depolymerizing activity [48]. NMR and mass spectrometry analysis determined that the commercial grade rigosertib contained at least 5% of ON-01500. This ON-01500 impurity may result from the degradation of rigosertib due to poor storage conditions, including elevated temperature, acidic pH, and exposure to intense light. In contrast, pharmaceutical-grade rigosertib directly supplied from Onconova Therapeutics (Onconova Therapeutics, Inc., Newtown, PA 18940, USA), which is free of the ON-01500 contaminant, does not exhibit in vitro tubulin-binding activity. More importantly, they also showed that different cell lines harboring the L240F mutations were sensitive to the pharmaceutical-grade rigosertib and failed to proliferate. The disagreement does not end here, as Jost et al. reevaluated the effect of commercial and clinical-grade rigosertib, showing indistinguishable phenotypes and agreeing with their previous findings [49]. Both rigosertib versions destabilize microtubules either in vivo or in vitro, and both show significantly reduced toxicity in cell lines expressing the L240F tubulin mutant. Interestingly, a very recent screen for identifying tubulin polymerization inhibitors pinpointed rigosertib as a major hit and demonstrated the capacity of rigosertib binding to tubulin by in silico docking analysis [50].

On the other hand, earlier studies made these conclusions even more controversial by showing that rigosertib-mediated effects do not depend on altered microtubule dynamics. Oussenko et al. not only described rigosertib-induced mitotic arrest as mentioned above, but also shed light on the microtubule destabilizer hypothesis by testing the ability of rigosertib and the inactive analog molecule ON-01911 to modify the tubulin polymerization. They revealed that clinical-grade rigosertib directly obtained from Onconova Therapeutics and control ON-01911 had a minor inhibition effect on tubulin polymerization, whereas nocodazole (another well-known microtubule poison) completely inhibited the process. This is indeed in agreement with the original article describing rigosertib as a Plk1 inhibitor, in which; they also established that rigosertib does not alter microtubule polymerization [1].

In summary, it is clear that rigosertib induces mitotic arrest and this appears to be a major cause of cell death. Whether this mitotic arrest is a direct effect of rigosertib due to changes in microtubule dynamics or other targets, such as Plk1, is still under debate. The source of the drug also appears to determine the outcome. While the clinical-grade rigosertib supplied by the proprietary pharmaceutical company Onconova Therapeutics appears to be more reliable than the commercial stocks, it is probably more important that the storage and handling conditions are precise to avoid degradation to synthesis intermediates, such as ON-01500, which has a strong affinity for microtubule heterodimers.

## 4. Rigosertib Clinical Trials for Cancer Therapy

Successful preclinical in vivo studies performed with rigosertib confirmed its safety profile with no evidence of myelotoxicity, neuropathy, or cardiotoxicity in xenograft mouse models, while successfully inhibiting liver, breast, and pancreatic cancers [1,51]. Rigosertib was, therefore, immediately included in a series of clinical trials to test its efficacy in a wide range of different cancers, including both solid tumors and hematological malignancies (Table 1).

Phase I trials suggested a maximum tolerated drug dose of 1800 mg/24 h in a three-day constant intravenous infusion rate, delivered every other week, with an acceptable toxicity profile [52,53,54]. The most common adverse events (AEs) included fatigue, gastrointestinal, and urinary symptoms. More severe grade 3 to 4 AEs were uncommon. Thrombocytopenia AE was also observed, yet to a lesser extent. Interestingly, drug-induced myelosuppression was also infrequent, thus providing a therapeutic advantage for myelodysplastic syndrome patients, given their compromised bone marrow function.

Rigosertib can be also administrated orally with a positive clinical activity [55]. Dose-limiting toxicity was established at 700 mg twice daily (b.i.d) upon severe grade 3 AE appearance, and the recommended dose was identified as 560 mg b.i.d. Overall, oral rigosertib was well tolerated. The most significant grade 2–3 toxicities included mainly urinary symptoms, abdominal pain and diarrhea, fatigue, hypotension, and anorexia. These positive toxicity and tolerability data encouraged clinicians to move forward into more advanced clinical trial phases.

Although some phase II tests were performed on some solid tumors, such as non-small cell lung cancer adenocarcinoma, and a wide variety of squamous cell carcinomas, they were predominately focused on hematological malignancies and, more concretely, on myelodysplastic syndromes (MDS) and myeloid leukemias. Similarly, phase III trials were mainly done in myelodysplastic syndromes, and only a recent phase III trial was performed on pancreatic adenocarcinoma patients (Table 1). It is also important to note that although most trials with rigosertib have been conducted as a monotherapy strategy (mainly phase I), advanced phase II and III studies have also been undertaken in combination with other chemotherapeutic agents in an effort to achieve a better response for those patients. Gemcitabine (DNA synthesis inhibitor) was combined with rigosertib in the phase III trial for pancreatic adenocarcinoma [8,56]. Azacitidine or decitabine (DNA methyltransferase inhibitors) were combined with rigosertib for MDS trials. Some solid tumor trials also combined rigosertib with either oxaliplatin (DNA synthesis inhibitor) or irinotecan (topoisomerase I inhibitor).

Despite the motivating number of clinical trials annotated in the U.S. National Library of Medicine website (ClinicalTrials.gov, accessed on 20 February 2023), very few of them provide results that indicate the efficacy of the drug. Data from phase I and II trials in MDS and myeloid leukemias were promising, with some patients showing a positive response to the treatment, with either a partial or complete bone marrow blast response and subsequent hematological improvement. This encouraged the initiation of a large-scale phase III trial comparing rigosertib administration to the standard best supportive care (BSC) for MDS (low dose cytarabine). Unfortunately, the results of the large trial were not positive [7]. The median overall survival (OS) in the rigosertib arm was 8.2 months and the median OS in the BSC arm was 5.9 months. In addition, no patients had a complete or partial response, although there is a modest increase in the bone marrow blast response with rigosertib, which was previously associated with improved survival. 

In summary, rigosertib did not reach its primary objective of providing a survival benefit compared to the BSC-treated cohort, although certain subgroups appeared to benefit from treatment. Another interesting phase III trial is the combination of rigosertib with gemcitabine in a cohort of metastatic pancreatic adenocarcinoma, with 160 enrolled patients [8,56]. Unfortunately, despite the good data from preclinical studies using PDX models of pancreatic tumors [51], the combination treatment did not show an improvement in survival or response in pancreatic cancer patients and the authors stated that further studies with rigosertib in this type of cancer would not be performed.

In an attempt to understand the variability in sensitivity to rigosertib, and to shed light on what may be the best tumoral context for a rigosertib therapeutic strategy, we took advantage of the DepMap portal (depmap.org), which provides chemical and genetic cancer dependency information, allowing data visualization from a large scale of viability screens across hundreds of cell lines. We selected data generated by the Broad Institute CTD^2 Center originally published via the Cancer Therapeutics Response Portal (CTRP) [57,58] to plot rigosertib sensitivity (AUC, area under the curve) against different cancer lineages. We observed a specific sensitivity against rigosertib of hematological cancers cells (leukemias, myeloid and lymphoid) in comparison with solid cancer-derived cell lines (Figure 3). This probably explains why clinical trials have focused primarily on myeloid and lymphoid tumors, although in vitro data have never really translated into the clinical trials for these cancers. 

Interestingly, and according to DepMap portal data, the most sensitive cell populations are those derived from neuroblastoma tumors. Indeed, the sensitivity of rigosertib to neuroblastoma models, both in vitro and in vivo, has been reported by others [46,47,59], demonstrating that rigosertib may be a good therapeutic alternative for this tumoral type. Perhaps the fact that neuroblastoma is a pediatric tumor is one of the reasons why its testing in clinical trials has been negatively affected. The data from the DepMap portal also confirm why the clinical trial that was focused on pancreatic cancer was completely unsuccessful. The cell lines derived from pancreatic adenocarcinoma are the most refractive to rigosertib treatment, among the 20 cohorts tested, with values equivalent to those for biliary tract and kidney tumors.

**Table 1 pharmaceutics-15-01232-t001:** Summary of rigosertib clinical trials.

Phase	NCT Number	Treatment	Disease	Participants	References
Phase I	NCT01048619	ON-01910.Na	Myelodysplastic Syndrome	36	[55]
	NCT01168011	ON-01910.Na	Solid Tumor	68	
	NCT01125891	Gemcitabine or ON-01910.Na	Malignant Neoplasms Solid Tumors	39	[60]
	NCT00854646	ON-01910.Na	Acute Myelocytic Leukemia Acute Lymphocytic Leukemia Chronic Myelocytic Leukemia Chronic Lymphocytic Leukemia Myelodysplastic Syndromes	22	[61]
	NCT01538537	ON-01910.Na	Advanced Cancer Solid Tumors Cancer Neoplasms	29	[53]
	NCT01538563	ON-01910.Na	Solid Tumors Advanced Cancer Neoplasms	42	-
	NCT01165905	Gemcitabine or ON-01910.Na	Solid Tumor	10	-
	NCT00861783	Irinotecan + ON-01910.Na Oxaliplatin + ON-01910.Na	Hepatoma Advanced Solid Tumor	16	-
	NCT00861328	Irinotecan + ON-01910.Na Oxaliplatin + ON-01910.Na	Advanced Solid Tumors	18	-
	NCT00861510	ON-01910.Na	Lymphoma Mantle-cell Leukemia Lymphocytic Chronic B-Cell Leukemia Hairy Cell Waldenstrom Macroglobulinemia Multiple Myeloma	16	[62]
	NCT00533416	ON-01910.Na	Myelodysplastic Syndrome (MDS)	14	[63]
Phase I/II	NCT01167166	ON-01910.Na	Acute Myelocytic Leukemia Acute Lymphocytic Leukemia Myeloproliferative Disease Chronic Myeloid Leukemia	30	-
	NCT00854945	ON-01910.Na	Myelodysplastic Syndromes Acute Myeloid Leukemia	36	-
	NCT01926587	Azacitidine + ON-01910.Na	Myelodysplastic Syndromes Acute Myeloid Leukemia Chronic Myelomonocytic Leukemia	45	[64]
	NCT04263090	Nivolumab + ON-01910.Na	Non-small Cell Lung Cancer Adenocarcinoma Stage IV	20	-
	NCT03786237	ON-01910.Na	Epidermolysis Bullosa Dystrophica Squamous Cell Carcinoma	12	-
Phase II	NCT01807546	ON-01910.Na	Head and Neck Squamous Cell Carcinoma Anal Squamous Cell Carcinoma Lung Squamous Cell Carcinoma Cervical Squamous Cell Carcinoma Esophageal Squamous Cell Carcinoma Skin Squamous Cell Carcinoma Penile Squamous Cell Carcinoma	64	-
	NCT01904682	ON-01910.Na	Myelodysplastic Syndromes	45	-
	NCT01584531	ON-01910.Na	Myelodysplastic Syndromes MDS Trisomy 8	82	-
	NCT00906334	ON-01910.Na	Myelodysplastic Syndromes	14	[65]
Phase III	NCT01241500	ON-01910.Na	Myelodysplastic Syndromes MDS RAEB Chronic Myelomonocytic Leukemia	299	-
	NCT01928537	ON-01910.Na	Myelodysplastic Syndromes Refractory Anemia with Excess Blasts Chronic Myelomonocytic Leukemia Cytopenia	67	-
	NCT02562443	Best supportive care (BSC) + ON-01910.Na Vs. Physician’s choice (PC) + ON-01910.Na	Myelodysplastic Syndromes MDS Refractory Anemia with Excess Blasts RAEB	372	[7]
	NCT01360853	Gemcitabine + ON-01910.Na	Metastatic Pancreatic Adenocarcinoma	160	[8,56,66]

## 5. Genetic Biomarkers Associated with Rigosertib Response

In an effort to elucidate the mechanism of action of rigosertib and to identify therapeutic biomarkers, some researchers have recently begun to identify the genetic alteration that may determine the efficacy of the response to rigosertib.

Oussenko et al. performed cross-resistance experiments and showed that rigosertib did not induce an MDR1 (multi-drug resistance 1) or an MRP1 (multi-drug resistance-associated protein 1) response. Moreover, cells resistant to other drugs did not show cross-resistance to rigosertib, making it a favorable anticancer agent [23]; hence, rigosertib appears to escape the general resistance mechanisms described for other drugs. On the contrary, Sánchez-Burgos et al. described how deficiency of the FBXW7 gene leads to resistance to rigosertib treatment. Interestingly, FBWX7 deficiency also confers resistance to a wide range of other chemotherapeutic agents, making it a general drug-resistance mechanism, rather than rigosertib-specific [67].

Another described biomarker of rigosertib efficacy is the deubiquitinating enzyme USP28 [68]. USP28 destabilizes the Raf family proteins by a feedback-loop-mediated mechanism. When USP28 is downregulated, B-Raf is hyper stabilized and, therefore, enhances downstream MAPK signaling, promoting resistance to Raf inhibitor therapy strategies. As USP28 is frequently deleted in a proportion of melanoma patients, the authors suggested rigosertib as a potential therapeutic strategy for USP28-depleted cells, demonstrating an increased sensitivity of USP28-depleted cells to rigosertib. Surprisingly, rigosertib appears to kill USP28 null cells in an ERK independent manner, and with a strong mitotic arrest phenotype. Hence, there is still doubt as to whether this USP28-null synthetic lethality is due to an amplification of the Ras–Raf signaling pathway, or simply to an increase in the proliferation ratio and a strong dependence on mitosis. Indeed, the same authors observed that USP28-depleted cells are more sensitive to small molecules such as microtubule poisons (taxol, vincristine, nocodazole) or even Plk1 inhibitors such as BI-2536.

Focusing on myelodysplastic syndromes (MDS), a possible biomarker for rigosertib response is the presence of trisomy 8. Trisomy of chromosome 8 and monosomy of chromosome 7 are common alterations observed in MDS patients [69] and these cytogenetic features correlate with the upregulation of the oncogenes c-myc, Wilms Tumor 1 (WT1), and Cyclin D1, thus increasing the proliferation capacity of these cells [70]. Interestingly, rigosertib has a selective inhibitory effect in trisomy-8 MDS cells and, to a lesser extent, in monosomy-7. This was also seen in patients enrolled in the MDS clinical trial (NCT00533416) and occurred even when these chromosomal abnormalities were part of a more complex karyotype. [63]. Thus, trisomy-8/monosomy-7 and elevated Cyclin D1, can behave as bona fide biomarkers for rigosertib treatment, at least in MDS.

Seeking to identify relevant cellular biomarkers following rigosertib treatment, Jost et al. performed a chemical screen for rigosertib using a CRISPRi (inactivation) and a CRISPRa (activation) strategy and identified a signature of microtubule-related genes that antagonized or synergized with the drug [5]. Specifically, the authors determined that the overexpression of the KIF2C gene, which encodes the microtubule depolymerizing enzyme MCAK [71], sensitizes cells to rigosertib. At the same time, the knockdown of TACC3, a microtubule-binding protein involved in microtubule stability [72], sensitizes cells to rigosertib. Thus, genetic manipulations that destabilize microtubules lead to increased sensitivity to rigosertib, whereas genetic alterations that induce microtubule stabilization lead to a protective effect against rigosertib [5]. To validate this hypothesis, the authors knocked-down both genes involved in microtubule stability with CRISPR-Cas9 technology and confirmed in different tumoral cell lines that the downregulation of KIF2C and TACC3 dictate resistance or sensitivity to rigosertib, respectively. In addition, they confirmed that rigosertib destabilizes microtubules by docking into the colchicine-binding site of tubulin. These results, although requiring further validation, identify specific genes whose expression predicts response to rigosertib and may serve as biomarkers for rigosertib treatment in a clinical context.

Despite these initial efforts to identify biomarkers of response to rigosertib, there is still very little known, and further efforts are needed in order to find genetic signatures that might help to better stratify patients in future clinical trials.

## 6. Conclusions, Current Challenges, and Prospects

Rigosertib is considered a promising therapeutic strategy for cancer for several reasons. It is defined as a non-ATP competitor multi-kinase inhibitor, which makes it an attractive molecule, as it avoids the associated drug-resistance mechanisms that can arise with ATP-competitor molecules [9]. Data suggest that rigosertib does not present cross-resistance to other drugs and is not an MDR1 or MPR1 substrate, making rigosertib a good choice of treatment for multi-drug resistant cancers. Preclinical studies prompted rigosertib as a promising molecule for the treatment of cancer therapy because in vivo studies demonstrated its efficacy in killing cancer cells with little associated toxicity. This tolerable toxicity was also seen in many clinical trials performed; therefore, rigosertib was considered a promising therapeutic strategy. Unfortunately, advanced clinical trials turned out to not be as efficient as initially thought and there was very little improvement when compared to the standard therapies for several types of cancer. This indicates that we still lack the right biomarkers to better stratify the patients for a rigosertib-mediated therapy. In this regard, all efforts trying to define the drug target have been confusing and controversial.

Rigosertib’s mechanism of action was originally suggested to be an allosteric inhibitor of the mitotic master regulator Plk1 [1]. Over the years, Plk1 specificity has been debated [22] and, in parallel, there was increasing evidence showing that rigosertib might alter other pathways. Indeed, it was described as a the PI3K–Akt axis inhibitor [2,3] or acting as a Ras–Raf binding mimetic leading to MEK signaling inhibition [4]. In the same vein, the inhibition of MEK signaling was attributed to the generation of mitotic stress [6], thus altering stress-signaling pathways. Finally, the mitotic arrest phenotype was recently explained by demonstrating that rigosertib is a microtubule-destabilizing drug [6]. This raised a strong controversy about the source quality of the drug, its stability, and its storage accuracy [48,49].

Because of the multiple mechanisms of action described, the scientific community has followed waves of studies, often viewing rigosertib as a unique inhibitor for Plk1, Ras, or PI3K/Akt, overlooking all other possible targets and side effects. Although some research groups have begun efforts to define the mechanisms of rigosertib, these studies are still at an early stage, and only a few genetic determinants have been described, with little clinical impact.

Therefore, a strong effort to elucidate rigosertib’s mechanism of action and understand how rigosertib inhibits cancer cell growth is critical to better define the potential patients who could benefit from this inhibitor and, thus, advance clinical trials.

## Figures and Tables

**Figure 1 pharmaceutics-15-01232-f001:**
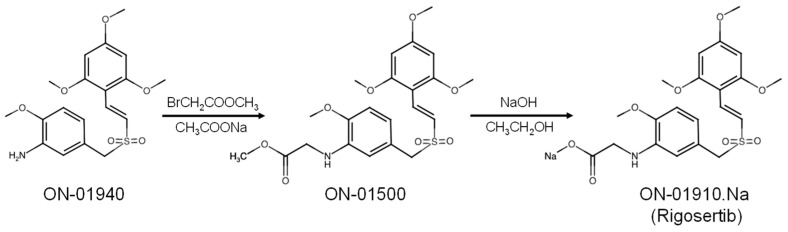
Scheme of development of the rigosertib molecule. Images generated using chemaxon.com/marvin (accessed on 25 January 2023).

**Figure 2 pharmaceutics-15-01232-f002:**
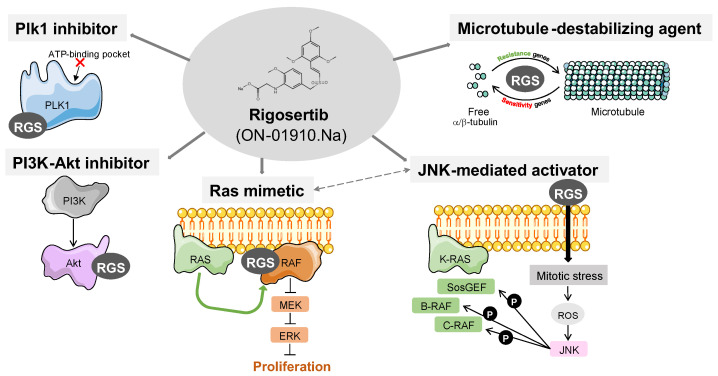
Diagram showing the described rigosertib targets and mechanisms of action. Rigosertib was first described as an allosteric Plk1 inhibitor [1]. Other data show that rigosertib can inhibit the PI3K–Akt signaling axis [2,3]. Rigosertib is also defined as a Ras–Raf binding competitor, altering the Ras–MAPK signaling cascade [4]. This Ras–Raf signaling impairment might be due to altering the response of stress kinases such as JNK [6]. Finally, using CRISPR genome-wide screens, rigosertib is also identified as a microtubule destabilizing agent [5].

**Figure 3 pharmaceutics-15-01232-f003:**
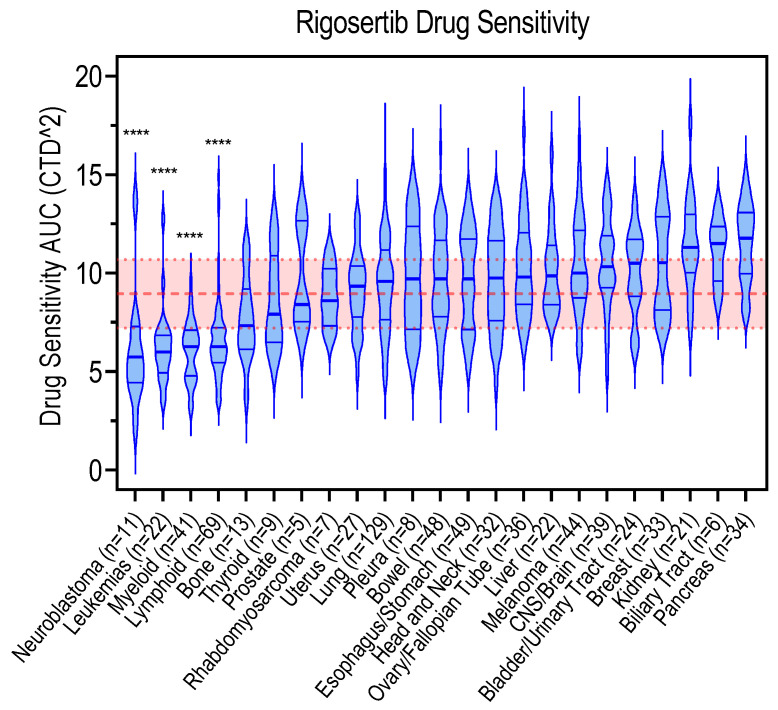
Data adapted from the DepMap Portal, retrieved from the Cancer Therapeutics Response Portal (CTRP), showing the sensitivity of cancer-cell lines of multiple origins to rigosertib. AUC reflects the area under the concentration-response curve. The lower the AUC, the more sensitive the cell line. Violin plots show median rigosertib sensitivity values (thick blue line). The thick dashed red line shows the average of all median AUCs. The shaded red area is defined by the standard deviation of the average median values. An unpaired t-test was performed, comparing the neuroblastoma, leukemias, myeloid, or lymphoid cohorts with the rest of all cohorts together and significance is indicated by *p* < 0.0001 (****). The number of cell lines (n) in each cohort is indicated in the X axis. Statistics were calculated using GraphPad Software version 9.5.1.

## Data Availability

Table 1 was generated retrieving data from the Clinical Trial web page (https://clinicaltrials.gov/ct2/home, accessed on 20 February 2023) of the U.S. National Library of Medicine. Figure 1 was generated using the chemical drawing tool Marvin JS (https://chemaxon.com/marvin, accessed on 25 January 2023). Figure 2 was drawn using templates from Servier Medical Art. Servier Medical Art by Servier is licensed under a Creative Commons Attribution 3.0 Unported License (https://creativecommons.org/licenses/by/3.0/, accessed on 15 January 2023). Figure 3 was generated using data from the DepMap Portal (https://depmap.org/portal/, accessed on 15 February 2023), retrieved from the Cancer Therapeutics Response Portal (CTRP).

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
