# Peer review of "Lights and Shadows on the Cancer Multi-Target Inhibitor Rigosertib (ON-01910.Na)"

_pharmaceutics, 2023, doi:10.3390/pharmaceutics15041232_

Round 1
Reviewer 1 Report
Rigosertib is a small molecule currently in phase III clinical trials for several myelodysplastic syndromes and leukemias and is therefore close to clinical translation. Rigosertib clinical progress has been hampered by the lack of understanding in its mechanism of action, being considered a multi-target inhibitor in the last years. Understanding the mechanism of action of rigosertib has potential clinical implications worth exploring, as it may help to tailor cancer therapies and improve patient outcomes.
First , rigosertib was described as an inhibitor of the mitotic master regulator Polo-like kinase 1 (Plk1). But the review highlights recent studies that have shown that rigosertib may also interact with the PI3K/Akt pathway, act as a Ras-Raf binding mimetic (altering the Ras signaling pathway), as a microtubule destabilizing agent, or as an activator of a stress-induced phospho-regulatory circuit that ultimately hyperphosphorylates and inactivates Ras signaling effectors.
The review is very comprehensive, nicely summarizes the history of rigosertib from development to the tumor types and pathways affected by this small molecule. It also emphasizes the importance to further elucidate rigosertib's mechanisms of action and cancer cell growth inhibition, as this is critical to better define the potential patients who could benefit from this inhibitor and advance clinical trials.
Author Response
We thank the reviewer for his/her effort in revising the manuscript, and for the positive comments and output.
Reviewer 2 Report
Title: Lights and shadows on the cancer multi-target inhibitor Rigosertib (ON-01910.Na)
This work mainly describes the mechanism involved in Rigosertib (ON-01910.Na) as the cancer multi-target inhibitor. From its development, related targets, clinical trials and Genetic biomarkers, Rigosertib (ON-01910.Na) was described in detail. Targets involved in Rigosertib (ON-01910.Na) were discussed in detail, including Plk1, PI3K-Akt, Ras, Ras-Raf-MEK and tubulin. The author describes that Rigosertib (ON-01910.Na) is a multi-target drug from the relevant literature. The author also analyzed the advantages and disadvantages of Rigosertib (ON-01910.Na) as a clinical drug in the description of targets and introduction of clinical trial results. This has potential clinical implications for understanding the mechanism of action of rigosertib, as well as helping to tailor cancer therapies and better define the potential patients who could benefit from this inhibitor and thus advance clinical trials. On the whole, the manuscript is rich and interesting, and the overall structure is clear, and this is a topic of interest for researchers in relevant fields, but there are still some small problems that need to be improved when the paper is published on Pharmaceutics. My detailed comments are as follows:
1. The first and second paragraphs of the introduction both introduce the pathways involved in Rigosertib. It is suggested to integrate together to reflect the relevance of the content.
2. The structural formula in Figure 1 is not very formal. It is recommended to modify it.
3. The "50" of "IC50" shall be located in the subscript (" 3.1 "section; Line 4);" The "50" of GI50" shall be located in the subscript (" 3.1 "section; Line 8).
4. The summary of “Rigosertib as a Plk1 inhibitor” is too simple. It is suggested to further explore the mechanism of this drug as a Plk1 inhibitor.
5. Where there is no reference in Table 1, it is suggested to use "\" or "-" instead.
Author Response
We thank the reviewer for his/her effort in revising the manuscript, and for the excellent suggestions to upgrade the manuscript.
The following are the answers to the suggested points:
1.- The reviewer is right regarding the introduction’s first paragraphs. We integrated both paragraphs into a single one, with the following result:
Rigosertib (ON-01910.Na) is a non-cyclooxygenase inhibitor sulfone metabolite with the ability to act as a non-ATP competitor kinase inhibitor. Originally described as an inhibitor for the master mitotic kinase Polo-like kinase (Plk1) [1], rigosertib has been reported to have broad activity against a number of different targets, some of which include kinases such as PI3K/Akt or NF-kB complex proteins [2,3]. Interestingly, it has also been described as a Ras mimetic molecule, inhibiting the Ras-Raf binding, and therefore hampering the Ras mitogenic signaling [4]. Recent data from CRISPR genome-wide screens, describe rigosertib as a microtubule-depolymerizing agent, leading to cell cycle alterations typical of other microtubule dynamic poisons [5]. It has also been proposed that the induced mitotic stress, upon rigosertib treatment, leads to oxidative stress resulting in a JNK-mediated inhibition of the Ras-MAPK pathway [6]. Overall, rigosertib is highly effective in killing tumoral cells in vitro and in vivo, being cells arrested in mitosis the most prominent phenotype, although there is abundant data showing that rigosertib effectively stops cell proliferation through various combined effects.
2.- For plotting Figure 2, we used the “chemaxon.com/marvin” web page for chemical molecule designing. We used the PubChem platform to get all chemical info on rigosertib and precursor molecules (https://pubchem.ncbi.nlm.nih.gov/compound/Rigosertib). Following the reviewer suggestion, we have modified figure 2 for a more formal structure, eliminating the colored residues and the point on the methyl groups.
3.- We made the formatting changes as suggested.
4.- We thank the reviewer for this comment and agree that the explanation for the Plk1 mechanism of rigosertib is somehow simple. Indeed, there is not much more information in the literature that the Gumireddy et al 2005 article, where they first describe the Plk1 inhibition effect. Since then, rigosertib was standardized as a Plk1 inhibitor and is often used as such, with no further evaluation. We again checked the literature and the only study we could find in which authors study more in depth the Plk1 phenotype is “Therapeutic Targeting PLK1 by ON-01910 Is Effective in Local Treatment of Retinoblastoma” by Ma et al., 2021; a paper that was originally cited in our review but not explained in detail. Yet, we have amplified the information in this paragraph for better understanding of future readers including a short summary of the study mentioned above (Ma et al., 2021). We included the following description:
To further evaluate the specificity of rigosertib for Plk1, the authors evaluated the activity of a panel of 29 recombinant kinases and showed that rigosertib was most active against Plk1 and therefore likely to be its primary target. However, rigosertib also showed some activity against other kinases such as PDGF receptor (PDGFR), Abl, Flt-1, CDK1, Plk2, Src when tested in recombinant kinase assays. Similarly, when the authors performed cell synchronization assays and tested for Plk1 activation, they showed that rigosertib inhibited Plk1 activity, although total levels of the kinase were not affected. This effect of rigosertib seemed to be specific for Plk1, since there is almost no alteration in CDK1 activity, although other kinases were not tested in these experiments. Since Plk1 is required for tumor cell proliferation, the authors show that rigosertib induced apoptotic activity against 94 tumor cell lines with a GI50 between 50-200 nM, resulting in tumor cell death by mitotic arrest, demonstrating its potential efficacy. Notably, this cytotoxic effect is not observed in non-tumoral cells. More recently, in an effort to evaluate the efficacy of rigosertib in retinoblastoma-derived cells, Huan Ma et al, showed suppression of Cdc25C phosphorylation, as a readout for Plk1 activity, and concomitant accumulation of CDK1 phosphorylation [26]. They also performed flow cytometry cell cycle analysis and demonstrated that upon rigosertib treatment, there was an arrest in the G2/M phase of the cell cycle, although not as strong as other reports.
5.- We made the formatting changes as suggested.
Reviewer 3 Report
The review is comprehensive and gives a good discussion about the struggles to get Rigosertib into the clinic. It is clear Rigosertib works by causing mitotic arrest, but the mode of action is still elusive. The review is very nicely written and figures and tables are clear. The only suggestion I have is to provide a bit more information about figure 3. It is really interesting heamatogical lines are more sensitive. Is this significant (the legend says statistics was done, but I cannot see this in the figure), how many lines are included in the data etc. ?
Author Response
We agree with reviewer number 2, and apologize for this mistake. Indeed, we made the statistical analysis but we forgot to include it in the figure legend. Figure legend is now updated in the revised version, with the statistical data comparing each of the hematological cell line cohort (also the neuroblastoma cohort), with all the other cohorts together. The number of samples per cohort is included in the figure.
Updated Legend Figure 3:
Figure 3. Data adapted from the DepMap Portal, retrieved from the Cancer Therapeutics Response Portal (CTRP), showing the sensitivity of cancer cell lines of multiple origins to rigosertib. AUC reflects the Area Under the concentration-response Curve. The lower the AUC, the more sensitive the cell line. Violin plots show median rigosertib sensitivity values (thick blue line). The thick dashed red line shows the average of all median AUCs. The shaded red area is defined by the standard deviation of the average median values. An unpaired t-test was performed comparing the neuroblastoma, leukemias, myeloid or lymphoid cohorts versus the rest of all cohorts together and significance is indicated by p<0.0001 (****). The number of cell lines (n) in each cohort is indicated in the X-axis. Statistics were calculated using GraphPad Software (www.graphpad.com)
Reviewer 4 Report
This manuscript provides the comprehensive view of the emerging cancer drug, Rigosertib. This review covers up-to-date findings on the fairly comprehensive topics including its development, mechanism of action, target signaling molecules, and clinical trials. This manuscript is also very well organized. Therefore, I recommend a manuscript accepted without revision.
Author Response

(The authors gave the same response as above.)
